# DDGS-CT: Direction-Disentangled Gaussian Splatting for Realistic Volume Rendering

**Zhongpai Gao**\*    **Benjamin Planche**\*    **Meng Zheng**
**Xiao Chen**    **Terrence Chen**    **Ziyan Wu**
United Imaging Intelligence, Boston, MA
`{first.last}@uii-ai.com`

## Abstract

Digitally reconstructed radiographs (DRRs) are simulated 2D X-ray images generated from 3D CT volumes, widely used in preoperative settings but limited in intraoperative applications due to computational bottlenecks, especially for accurate but heavy physics-based Monte Carlo methods. While analytical DRR renderers offer greater efficiency, they overlook anisotropic X-ray image formation phenomena, such as Compton scattering. We present a novel approach that marries realistic physics-inspired X-ray simulation with efficient, differentiable DRR generation using 3D Gaussian splatting (3DGS). Our direction-disentangled 3DGS (DDGS) method separates the radiosity contribution into isotropic and direction-dependent components, approximating complex anisotropic interactions without intricate runtime simulations. Additionally, we adapt the 3DGS initialization to account for tomography data properties, enhancing accuracy and efficiency. Our method outperforms state-of-the-art techniques in image accuracy. Furthermore, our DDGS shows promise for intraoperative applications and inverse problems such as pose registration, delivering superior registration accuracy and runtime performance compared to analytical DRR methods.

## 1   Introduction

**Motivation.**   Digitally reconstructed radiographs (DRRs) are simulated (*in silico*) 2D X-ray images rendered from 3D computational tomography (CT) volumes. While DRRs are widely utilized in preoperative settings, such as optimizing dose delivery and in radiation oncology [20, 41, 6], their potential intraoperative applications remain underexplored due to computational bottlenecks and naive modeling capability [1, 27]. For instance, real-time multimodal registration for image-guided procedures is currently impractical because of the time-consuming process of generating DRRs and integrating them into slice-to-volume registration [36].

The advent of GPU-accelerated computing has significantly improved the efficiency of radiography simulators [3, 39, 5], though they still fall short of meeting the stringent requirements of real-time applications. Recent developments in inverse graphics have led to the creation of differentiable DRR renderers (*e.g.*, DiffDRR [11], X-Gaussian [7], GaSpCT [25]), which show promise for inverse problems such as 2D/3D CT image registration [12]. However, these methods do not fully capture the complexities of X-ray image formation. They typically use computationally efficient ray-tracing techniques that model the attenuated photon fluence at each detector pixel by accumulating Hounsfield unit (HU) values along the 3D view-ray through the voxelized CT volume. This approach, while fast, cannot account for complex noise-inducing physical effects such as Compton scattering and beam hardening [14], crucial for accurate X-ray imaging.

---

\*Equal contribution

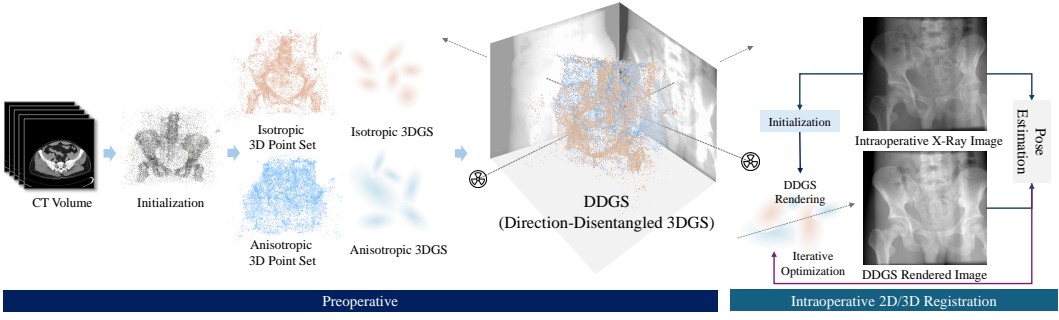

Figure 1: Proposed pipeline of Direction-Disentangled 3D Gaussian Splatting (DDGS).

Physics-based Monte-Carlo simulations [2, 15, 3, 4] do not suffer from these limitations. Relying on the decomposition of CT volumes into realistic materials via HU thresholding, they accurately simulate single-photon transport and probabilistically evaluate photon-matter interactions (photo-electric effect, Compton scattering, Rayleigh scattering, *etc.*) to determine the final attenuation and hit-region of each photon. This approach provides a more accurate representation of X-ray image formation; but the computational intensity of these methods—requiring $>10^8$ iterations to render an image—makes them impractical for real-time applications and inverse problems.

In this paper, we propose a novel, more elegant balance between (a) realistic physics-inspired X-ray simulation and (b) efficient, differentiable DRR generation, as illustrated in Figure 1.

**Contributions.** Our approach leverages 3D Gaussian splatting (3DGS) [17] for efficient DRR generation from CT volumes, similar to X-Gaussian [7] and GaSpCT [25]. However, we introduce a crucial adaptation: our direction-disentangled 3DGS (DDGS) method accounts for the dependencies of X-ray image formation on light directionality. Specifically, we decompose the radiosity contribution of 3D regions into two independent components: a non-directional component modeling the isotropic X-ray interactions with the corresponding material, and a direction-dependent component approximating higher-dimensional anisotropic interactions, such as scattering. Unlike the more generic view-dependent spherical-harmonics decomposition adopted by common 3DGS solutions [10], ours is tuned to the specific behaviors of high-energy photons. This formulation allows our method to implicitly learn and reproduce the effects of scattering on X-ray images without the need for complex photon-transport simulations at runtime.

Additionally, we adapt the initialization phase of 3DGS to account for the properties of tomography data. We adopt a geometry- and intensity-based dual strategy to sample 3D points that correspond to: (a) contact regions between different anatomical elements, where photon transport is likely to show discontinuities; and (b) homogeneous regions, where the primary attenuation contribution can be sparsely modeled but could benefit from view-dependent noise modeling.

We demonstrate that our solution achieves higher image accuracy compared to state-of-the-art methods across various medical benchmarks Additionally, we showcase the efficient application of our method to inverse problems, such as pose registration, highlighting its potential for real-time intraoperative use cases. Our DDGS achieves better registration accuracy and runtime performance than analytical DRR methods.

## 2   Related Work

The generation of DRRs and the simulation of X-ray images have been subjects of extensive research due to their critical roles in medical imaging, particularly in radiotherapy planning and diagnostic imaging [20, 41, 6, 43]. In this section, we provide some insight on X-ray imaging, traditional simulation tools, and the recent integration of machine-learning techniques.

**X-ray Image Simulation.**   X-ray imaging relies on the interactions of high-energy photons—emitted by the X-ray source of the scanner—with the matter (*e.g.*, body part) placed in between the X-ray source of the scanner and its radiation-sensitive detector cells (dexels). Imaging contrast arises from

differential absorption and scattering of X-ray photons by tissues, primarily through the photoelectric effect, Compton scattering, and, to a lesser extent, beam hardening for polychromatic scanners. The photoelectric effect involves the absorption of an X-ray photon by an atom and is more pronounced in higher atomic number materials like bone, causing them to appear whiter on X-ray images[2]; whereas Compton scattering occurs when an X-ray photon collides with an outer electron, resulting in the photon being deflected and losing energy, which contributes to image noise and reduced contrast [14, 29].

A variety of Monte Carlo (MC) simulation suites [2, 15, 3, 4] model these X-ray interactions and light transport. They trace the paths of a large number of individual photons as they undergo probabilistic interactions within the material. By simulating numerous photon trajectories, Monte Carlo methods account for the random nature of photoelectric absorption and Compton scattering, as well as secondary effects such as fluorescence and electron transport. These MC solutions produce highly realistic simulations of radiographic images, though at a high computational cost which makes them inadequate for most DRR use-cases. Furthermore, for these tools to render realistic DRRs from CT volumes, one must first pre-process the CT data to assign proper materials and their corresponding physical properties to each voxel, which is essential for realistic simulation. This step is often approximated by decomposing the volume into a small set of predefined materials according to user-provided HU ranges (*e.g.*, "*air*" material assigned to voxels with HU value $< -800$, "*lung*" assigned to voxels in range $] -800, -200]$, "*fat*" for $] -200, -100]$, *etc.*). This process can be time-consuming and prone to errors, affecting the fidelity of MC-based DRR generation [9].

**Efficient and Differentiable DRR Generation.**   Analytical DRR rendering methods [39, 38, 5] have been developed in parallel to the aforementioned MC suites for real-time or near-real-time applications. These methods have a much lower computational footprint, as they do not simulate individual photon interactions but instead use mathematical models to directly calculate the primary paths and approximated attenuations of X-rays. The seminal work of Siddon [31] on computing line integrals through a discretized CT volume is still at the core of most recent DRR solutions, *e.g.*, which proposed differentiable formulation of the rendering steps [11] or integrated neural networks to add realistic noise to the output [35]. However, the inherent approximations in these methods can affect their accuracy, particularly in modeling artifacts present in heterogeneous tissues or complex anatomical structures, *e.g.*, caused by scattering (non-primary light contributions).

Recently, researchers [7, 25] have tried to apply 3D Gaussian splatting (3DGS) techniques [17] to DRR rendering, *i.e.*, optimizing a cloud of 3D Gaussians to approximate voxel data, thereby enabling faster rendering with minimal accuracy loss. However, these early solutions overlooked the specific nuances of X-ray imaging, directly applying 3DGS methods designed for natural imaging [25] or oversimplifying the physical properties of X-ray attenuation [25], leading to suboptimal modeling.

In this paper, we propose a novel formulation of 3D Gaussian splatting, tuned to more efficiently approximate X-ray imaging (time- and quality-wise).

## 3   Methodology

### 3.1   Preliminaries

**Gaussian Splatting.**   3DGS [17] is a point-based rendering technique using 3D Gaussians to explicitly represent scenes in a more compact manner than volumetric representations. Each Gaussian $g_i = \{\boldsymbol{\mu}_i, \boldsymbol{\Sigma}_i, \boldsymbol{\alpha}_i, \boldsymbol{f}_i\}$ is defined by its mean position $\boldsymbol{\mu}_i \in \mathbb{R}^3$, covariance matrix $\boldsymbol{\Sigma}_i \in \mathbb{R}^{3 \times 3}$ (usually decomposed into separate scale and rotation parameters), opacity $\boldsymbol{\alpha}_i \in \mathbb{R}$, and radiance properties $\boldsymbol{f}_i \in \mathbb{R}^k$, where $k$ depends on the light contribution model adopted. *E.g.*, $k = (L+1)^2 \times 3$ for anisotropic, view-dependent RGB radiance decomposed into spherical harmonics (SH) of degree $L$, a model commonly used for natural-imaging applications. Images are rendered by casting view-rays through each pixel $p$ into the scene and alpha-blending the Gaussian contributions to the final ray color $C$, as:

$$C(p) = \sum_{j \in n} c_j \sigma_j \prod_{l=1}^{j-1} (1 - \sigma_l) \ \text{ with } \sigma_i = \boldsymbol{\alpha}_i e^{-\frac{1}{2}(p - \widehat{\boldsymbol{\mu}}_i)\widehat{\boldsymbol{\Sigma}}_i^{-1}(p - \widehat{\boldsymbol{\mu}}_i)}, \tag{1}$$

---

[2]X-ray film is historically white, turning black when exposed to high-energy photon.

where $n$ is the number of Gaussians considered, $c_j$ is the radiance (computed from $\boldsymbol{f}_j$, *e.g.*, via SH) of the $j$th Gaussian on the ray, and $\widehat{\boldsymbol{\mu}}_j$ and $\widehat{\boldsymbol{\Sigma}}_j$ are the image-plane projections of $\boldsymbol{\mu}_j$ and $\boldsymbol{\Sigma}_j$.

Since the aforementioned rasterization process is differentiable w.r.t. to the Gaussian parameters, the 3DGS representation of a scene can be learned via gradient-descent, given a set of 2D scene observations $\boldsymbol{I}_{\text{GT}}$ and their corresponding camera parameters. A combination of $\ell_1$ and SSIM [40] loss functions are commonly-adopted as criterion for this iterative optimization process [17]:

$$\mathcal{L} = (1 - \lambda)\ell_1(\boldsymbol{I}_{\text{GT}}, \boldsymbol{I}) + \lambda \operatorname{SSIM}(\boldsymbol{I}_{\text{GT}}, \boldsymbol{I}), \tag{2}$$

with $\boldsymbol{I}$ the rendered images (based on provided camera parameters), and $\lambda$ a loss-weighting hyper-parameter. The initial state is typically obtained by sampling relevant 3D points in the scene domain for the Gaussians, *e.g.*, from a structured-from-motion (SfM) point-cloud [34, 28].

**Application to DRR.** The volumetric rendering performed by 3DGS methods is conceptually similar to the one performed by analytical DRR methods [31, 11], *i.e.*, aggregating the attenuation values of light-rays cast through the voxelized CT volume. This similarity has recently motivated researchers [7, 25] to apply 3DGS models to DRR applications, driven by the compactness of 3DGS representations compared to voxel data (addressing the $\mathcal{O}(n^3)$ complexity inherent to voxel rendering). By optimizing a 3DGS model to approximate the visual properties of the CT volume, the resulting representation can render accurate DRRs much faster than traditional methods, making it more suitable for real-time *intra-operative* visualization. Consequently, the use of traditional, slower rendering solutions can be limited to generating DRR targets ($\boldsymbol{I}_{\text{GT}}$) for the offline (*i.e.*, *pre-operative*) training.

However, these early 3DGS-for-DRR solutions [7, 25] do not account for noise-inducing photon interactions (*e.g.*, scattering) when applying the analytical methods [30, 5] to render their training data. Since scatter-free DRRs are not affected by ray directions (*i.e.*, the attenuation at each point is naively considered isotropic), Cai et al. [7] simplified the radio-intensity function of each Gaussian to $c_i = \operatorname{sigmoid}(\boldsymbol{b} \cdot \boldsymbol{f}_i)$, where $\boldsymbol{b} \in \mathbb{R}^k$ is a direction-independent optimizable basis vector shared by all splats. Furthermore, their Gaussian point-cloud is initialized via evenly-spaced Cai et al. [7] or random-uniform [25] sampling of the scanned 3D space, *i.e.*, ignoring relevant geometrical and material properties of the target CT data. Inadequate initialization strategies negatively impact the convergence of 3DGS models and the resulting image accuracy [16]. In this work, we propose to reformulate both the initialization strategy and the radiance function of the 3D Gaussians to address these limitations.

## 3.2 Disentanglement of Isotropic and Anisotropic 3D Gaussians

We argue that, while X-ray scattering models are incompatible with the 3DGS rendering formulation (secondary rays would have to be cast, increasing the computational footprint exponentially), their anisotropic impact on X-ray imaging can be approximated to some extent by direction-dependent radiance functions. However, modeling this high-dimensional residual contribution is only meaningful if it appears in the training DRRs, *i.e.*, if the target radiographs are rendered using physics-inspired simulation tools. This is not always possible, *e.g.*, if the mapping from HU values to materials for the target CT scans is not provided. Ideally, in such cases, when only scatter-free DRRs can be generated, the radiance function should gracefully degrade to a lighter anisotropic formulation (*e.g.*, similar to [7]), to avoid unnecessary computations that could impact the model optimization, as well as its runtime performance. Moreover, X-ray interactions with matter themselves involve both isotropic (*e.g.*, photoelectric absorption/fluorescence, Rayleigh scattering) and anisotropic (*e.g.*, Compton scattering) phenomena [14, 22]. Therefore, for the sake of both modularity and accuracy, we propose to decompose our DRR representation into isotropic Gaussians $g_i^{\text{iso}}$ and anisotropic, direction-dependent ones $g_j^{\text{dir}}$. We formulate their respective radiosity functions as:

$$c_i^{\text{iso}} = \operatorname{sigmoid}(\boldsymbol{b}^{\text{iso}} \cdot \boldsymbol{f}_i^{\text{iso}}) \quad ; \quad c_j^{\text{dir}} = \operatorname{sigmoid}(Y_{1..L}(\theta, \phi) \cdot \boldsymbol{B}^{\text{dir}} \boldsymbol{f}_j^{\text{dir}}), \tag{3}$$

where $\boldsymbol{f}_i^{\text{iso}}, \boldsymbol{f}_j^{\text{dir}} \in \mathbb{R}^k$ are feature vectors respectively encoding the contribution of isotropic and anisotropic Gaussians, $\boldsymbol{b}^{\text{iso}} \in \mathbb{R}^k$ is a global isotropic basis vector (similar to [7]), and $\boldsymbol{B}^{\text{dir}} \in \mathbb{R}^{k_L \times k}$ is a second, anisotropic basis matrix that interacts with the $L$-degree spherical-decomposition $Y : \mathbb{R}^2 \mapsto \mathbb{R}^{k_L}$ applied to the ray angles $\theta, \phi$ without constant term (degree $= 0$), resulting in $k_L = L(L + 2)$.

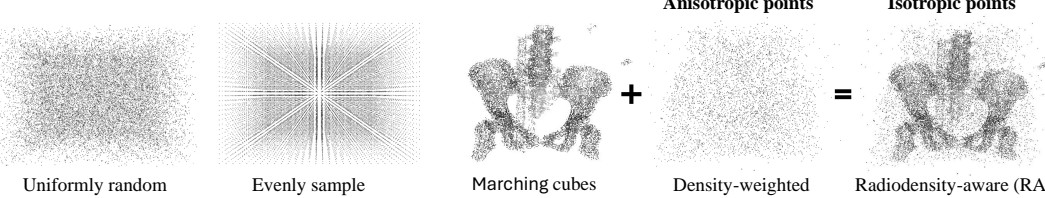

| Uniformly random | Evenly sample | Marching cubes | + | Density-weighted | = | Radiodensity-aware (RA) |

Figure 2: Illustration of the different sampling strategies for 3DGS initialization.

It is important to note that, while the usual spherical-harmonics representation for Gaussian splatting also contains isotropic (degree $= 0$) and anisotropic (degree $> 0$) components, the model assumes that both contributions are co-located (*i.e.*, they belong to the same 3D Gaussian $g_i$ of position $\boldsymbol{\mu}_i$ and covariance $\boldsymbol{\Sigma}_i$). To account for the complexity of X-ray light transport, we relax this co-location constraint, *i.e.*, modeling isotropic and anisotropic contributions via distinct 3D Gaussians ($g_i^{\text{iso}}$ and $g_j^{\text{dir}}$). We demonstrate in our evaluation that this disentanglement results in both higher image quality and lighter representation.

### 3.3 Initialization via Radiodensity-Aware Dual Sampling (RADS)

Our second main contribution targets the initialization of the point-cloud that supports the Gaussian-mixture modeling of the 3D data. Since 3DGS was defined for natural imaging (*e.g.*, ignoring sub-surface light transport), common initialization strategies limit their sampling to scene-surface points (SfM- [17] or depth-guided [24] subsampling). Such sampling of the 3D space is, however, inadequate for volumetric data, ignoring possibly salient regions. While uniform sampling of the CT space could be considered [7, 25], we argue that such a strategy is also suboptimal for anatomical data, discarding domain-relevant properties. We demonstrate empirically (*c.f.* Section 4) that uniform sampling results in slower convergence of the 3DGS representation and degraded runtime performance.

Therefore, we propose a novel twofold sampling strategy that accounts for the specific nature of CT data and for our dual isotropic/directional 3DGS model. Based on the radiodensity values contained in the target CT volumes, our solution samples two set of 3D points $\mathbf{P}^{\text{mc}} \in \mathbb{R}^{n_1 \times 3}$ and $\mathbf{P}^{\text{dw}} \in \mathbb{R}^{n_2 \times 3}$ with distinct, complementary distributions ($n_1, n_2$ scalar hyper-parameters).

Not unlike [24], the first set of points is sampled by applying the marching-cubes algorithm [21] to extract points at the interface between materials with different physical properties (*i.e.*, distinct HU responses in the CT volume). These interface regions typically need careful modeling when simulating X-ray imaging, as they result in discontinuous photon transport.

To complement this first set of points, the second set is semi-randomly sampled from the voxel centroids, according to a uniform distribution weighted by the voxels' radiodensities. *I.e.*, voxels with higher radiodensity, contributing to the X-ray attenuation, are more likely to be picked to initialize the Gaussians. In other words, this radiodensity-weighted strategy identifies additional points within the homogeneous regions of the CT volume that also contributes to the imaging process.

Based on their distinct properties, we assign the entire $\mathbf{P}^{\text{mc}}$ to initialize the isotropic Gaussians, whereas we equally split $\mathbf{P}^{\text{dw}}$ to initialize both isotropic and direction-dependent Gaussians. All in all, our dual radiodensity-aware strategy improves the initialization of the 3DGS representation and is more explainable than prior work (see Figure 2).

## 4 Experiments

### 4.1 Experimental Protocol

**Implementation.** For evaluation, we set the decomposition degree to $L$=1, the feature dimension to $k$=8, and initial cloud sizes to $n_1$=15,000 and $n_2$=10,000. We apply a loss weight $\lambda = 0.2$. We choose the default threshold level (*i.e.*, the average of the minimum and maximum volume) for the marching-cubes algorithm. DDGS training is performed on a single NVIDIA A100 using an Adam

Table 1: Comparison of Gaussian splatting-based DRR rendering techniques, on 2 datasets.

| | | 3DGS [17] ($L = 1$) | | | X-Gaussian [7] ($k = 32$) | | | DDGS (**Ours**, $L = 1, k = 8$) | | |
|---|---|---|---|---|---|---|---|---|---|---|
| | | # points↓ | PSNR↑ | SSIM↑ | # points↓ | PSNR↑ | SSIM↑ | # points↓ | PSNR↑ | SSIM↑ |
| NAF-CT | abdomen | 11,149 | 47.43 | 0.994 | **10,802** | 47.17 | 0.993 | 13,928 | **48.09** | 0.994 |
| | chest | 13,669 | 44.42 | 0.988 | 16,568 | 43.33 | 0.987 | **13,533** | **44.50** | 0.989 |
| | foot | **8,616** | 44.51 | 0.984 | 10,909 | 44.34 | 0.985 | 10,786 | **44.70** | 0.985 |
| | jaw | **17,902** | 40.47 | 0.973 | 22,318 | 40.02 | 0.972 | 19,665 | **40.57** | 0.974 |
| | avg | **12,834** | 44.21 | 0.985 | 15,149 | 43.72 | 0.984 | 14,478 | **44.47** | 0.986 |
| CTPelvic1K-Dataset6 | 001 | 53,988 | 35.40 | 0.971 | 50,059 | 36.81 | 0.979 | **41,947** | **37.88** | 0.984 |
| | 002 | 49,099 | 37.03 | 0.982 | 48,113 | 37.79 | 0.986 | **43,933** | **38.43** | 0.988 |
| | 003 | 60,755 | 35.73 | 0.973 | 59,822 | 36.46 | 0.977 | **48,536** | **38.28** | 0.983 |
| | 004 | 42,349 | 38.87 | 0.982 | **37,243** | 39.84 | 0.985 | 39,176 | **40.35** | 0.986 |
| | 005 | 42,482 | 39.37 | 0.984 | 46,014 | 39.30 | 0.984 | **39,570** | **40.36** | 0.987 |
| | 006 | 53,832 | 37.14 | 0.983 | 57,197 | 37.42 | 0.983 | **47,398** | **38.26** | 0.986 |
| | 007 | 45,360 | 37.09 | 0.980 | 48,454 | 37.62 | 0.983 | **41,032** | **38.87** | 0.987 |
| | 008 | 51,211 | 38.03 | 0.980 | 45,750 | 38.70 | 0.982 | **43,577** | **39.81** | 0.986 |
| | 009 | 44,060 | 38.19 | 0.981 | 41,279 | 38.92 | 0.985 | **41,389** | **39.87** | 0.987 |
| | 010 | **38,691** | 37.80 | 0.984 | 40,030 | 37.55 | 0.985 | 39,691 | **38.73** | 0.987 |
| | avg | 48,183 | 37.47 | 0.980 | 47,396 | 38.04 | 0.983 | **42,625** | **39.08** | 0.986 |

optimizer [18] with a learning rate of $1.25 \times 10^{-4}$ for $\boldsymbol{b}^{\text{iso}}$ and $\boldsymbol{B}^{\text{dir}}$ and $2.5 \times 10^{-3}$ for $\boldsymbol{f}^{\text{iso}}$ and $\boldsymbol{f}^{\text{dir}}$. Default 3DGS [17] learning rates are applied to the remaining parameters.

**Datasets.** We consider 4 datasets. (1) NAF-CT [42] includes four CT images of abdomen, chest, foot, and jaw. For each, we adopt TIGRE [5] to sample 50 evenly-distributed projections for training and 50 randomly-distributed ones for testing, in the range of $[-90°, 90°]$ (scatter-free DRRs). (2) CTPelvic1K [19] is a large dataset of pelvic CT images. We consider the first 10 scans of the sub-dataset6 in our experiments. We adopt DeepDRR [35] to generate 60 training (evenly-distributed) and 60 (randomly-distributed) testing DRRs (in the range of $[-60°, 60°]$), taking advantage of the scattering-modeling capability of DeepDRR to render realistic X-ray data. (3) Ljubljana [26] is a clinical dataset of 10 patients undergoing neurovascular surgery. Each patient underwent one CT and two X-ray angiography scans. The pose of each X-ray is also provided. Following DiffPose strategy [12], we randomly sample 900 projections for training and 100 for testing. (4) Finally, we provide some additional qualitative results on DeepFluoro [13], a collection of pelvic X-rays and CT images from six cadavers.

**Metrics.** We evaluate our method in two settings: novel-view synthesis and intraoperative 2D/3D image registration (*i.e.*, pose estimation). For novel-view synthesis, we use the standard PSNR and SSIM as metrics. For intraoperative 2D/3D image registration, we consider the error both in terms of rotation (angular distance) and translation (Euclidean distance). We further measure the clinically-relevant registration accuracy of key anatomical landmarks for Ljubljana scans [26] where they are provided. Following [12], we compute the target registration error (TRE) w.r.t. the positioning of these landmarks after registration.

### 4.2 Novel-View Synthesis

We first validate our contributions in terms of the quality of rendered DRRs and compactness of our dual Gaussian-mixture representation.

**Comparative Evaluation.** We compare to 3DGS [17] and X-Gaussian [7] on the NAF-CT [42] and CTPelvic1K [19] datasets for the novel-view synthesis. For each method, we adopt the hyper-parameters recommended by their respective authors (*e.g.*, $L = 1$ and 3 for normal 3DGS [17] and $k = 8$ and 32 for X-Gaussian [7]).

As shown in Table 1, for NAF-CT [42] (without scatter-related noise), our DDGS outperforms both 3DGS and X-Gaussian in terms of PSNR and SSIM. For CTPelvic1K [19] (with scatter), our DDGS significantly outperforms 3DGS and X-Gaussian in both PSNR and SSIM while using considerably

Table 2: Novel-view synthesis evaluation on CTPelvic1K data.

| | | Iteration=500 | | 2000 | | 7000 | | 15000 | | 30000 | |
|---|---|---|---|---|---|---|---|---|---|---|---|
| | | PSNR | SSIM | PSNR | SSIM | PSNR | SSIM | PSNR | SSIM | PSNR | SSIM |
| DDGS (**Ours**) | 001 | 25.77 | 0.908 | 30.46 | 0.944 | 35.18 | 0.975 | 36.91 | 0.981 | 38.04 | 0.984 |
| ($L=1, k=8$) | 002 | 27.64 | 0.921 | 31.60 | 0.954 | 35.33 | 0.977 | 36.90 | 0.984 | 38.30 | 0.987 |
| DDGS (**Ours**) | 001 | 26.20 | **0.909** | **31.50** | **0.950** | **35.65** | **0.978** | **37.68** | **0.983** | **38.15** | **0.985** |
| ($L=3, k=8$) | 002 | 27.87 | **0.923** | 32.23 | 0.956 | **36.02** | 0.979 | **37.87** | **0.986** | **38.97** | **0.989** |
| 3DGS [17] | 001 | 23.56 | 0.873 | 27.52 | 0.914 | 31.66 | 0.951 | 33.64 | 0.963 | 35.36 | 0.970 |
| ($L=1$) | 002 | 24.55 | 0.891 | 29.17 | 0.932 | 32.82 | 0.965 | 35.33 | 0.977 | 37.25 | 0.982 |
| 3DGS [17] | 001 | 23.55 | 0.872 | 27.54 | 0.914 | 33.34 | 0.962 | 36.01 | 0.975 | 37.12 | 0.980 |
| ($L=3$) | 002 | 24.55 | 0.891 | 28.96 | 0.931 | 34.50 | 0.973 | 37.72 | 0.985 | 38.75 | 0.988 |
| X-Gaussian [7] | 001 | 17.31 | 0.794 | 24.79 | 0.877 | 29.00 | 0.923 | 30.72 | 0.933 | 33.39 | 0.947 |
| ($k=8$) | 002 | 13.45 | 0.742 | 27.19 | 0.904 | 32.72 | 0.962 | 34.52 | 0.973 | 36.04 | 0.978 |
| X-Gaussian [7] | 001 | **26.34** | 0.854 | 31.28 | 0.950 | 33.56 | 0.968 | 35.07 | 0.974 | 36.83 | 0.979 |
| ($k=32$) | 002 | **28.32** | 0.868 | **33.44** | **0.966** | 35.85 | **0.980** | 36.55 | 0.984 | 37.89 | 0.986 |

Table 3: Ablation study w.r.t. the proposed disentangled isotropic/anisotropic representations.

| | DDGS (**Ours**) | | direct-entangled | | direct-independent | | direct-dependent | | 3DGS-disentangled | |
|---|---|---|---|---|---|---|---|---|---|---|
| | PSNR | SSIM | PSNR | SSIM | PSNR | SSIM | PSNR | SSIM | PSNR | SSIM |
| 001 | **38.04** | **0.984** | 36.17 | 0.977 | 35.93 | 0.976 | 32.48 | 0.942 | 35.32 | 0.971 |
| 002 | **38.30** | **0.987** | 37.32 | 0.984 | 36.96 | 0.983 | 33.04 | 0.953 | 35.38 | 0.973 |

fewer points. Therefore, our DDGS method is more effective in simulating realistic X-ray images from CT scans.

Furthermore, we compare our DDGS (with degrees $L=1$ and $L=3$ and feature dimension $k=8$) with traditional 3DGS (degrees $L=1$ and $L=3$) and X-Gaussian (feature dimensions $k=8$ and $k=32$) on the 001 and 002 data of CTPelvic1K [19] at iterations of 500, 2000, 7000, 15,000, and 30,000, as shown in Table 2. Our model with $L=1$ and $k=8$ performs comparably to 3DGS with $L=3$ and outperforms X-Gaussian with $k=32$ after 15,000 iterations. Experiments on more data can be found in Table S1. Our model with $L=3$ achieves the best performance overall. Figure 3 presents a qualitative comparison of the testing set, illustrating both the synthetic views and the generated 3D Gaussian points.

It should also be highlighted that, even though both the original CT scan (as voxel grid) and our 3DGS-based solution are explicit representations, the latter is significantly more compact. *E.g.*, our model can represent a CTPelvic1K scan with only 42,625 Gaussians (*c.f.* Table 1), defined by 19 float values each (3D position/rotation + 3D covariance + opacity + feature vector) and a shared 32-dimensional basis vector **b**, so 809,907 float values in total; whereas the original CT scans are each composed of $512 \times 512 \times 500 = 93,363,200$ values. Indeed, the voxel data may contain large homogeneous regions, which can be approximated with few Gaussians, hence a significant compression rate.

Finally, similar to [12], we qualitatively compare the results of recent DRR methods with actual X-ray images, for datasets providing both CT scans and real, posed projections (DeepFluoro [13], Ljubljana [26]). Results are shared in Figure 4, with our method achieving state-of-the-art PSNR. Further analyses are provided in Appendix B.

**Ablation Study.** Table 3 demonstrates the effectiveness of our direction-disentangled representation for 3D Gaussians. For several settings, we observe performance degradation when learning the direction-dependent and direction-independent components in the same 3D Gaussians (*direct-entangled*), learning the direction-independent 3D Gaussians alone (*direct-independent*), or learning the direction-dependent 3D Gaussians alone (*direct-dependent*). Additionally, we decouple the direction-dependent and direction-independent components in 3DGS [17] (*3DGS-disentangled*), which also showed lower performance compared to our learnable features. Figure 5 shows the rendered views from disentangled 3D Gaussians, highlighting that the direction-dependent 3D Gaussians

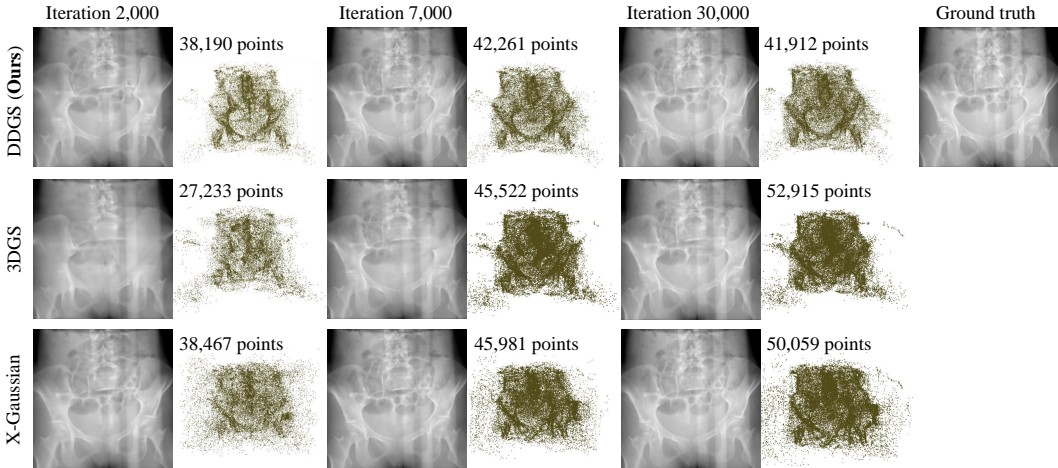

Figure 3: Visualization of the Gaussian cloud optimization for different methods.

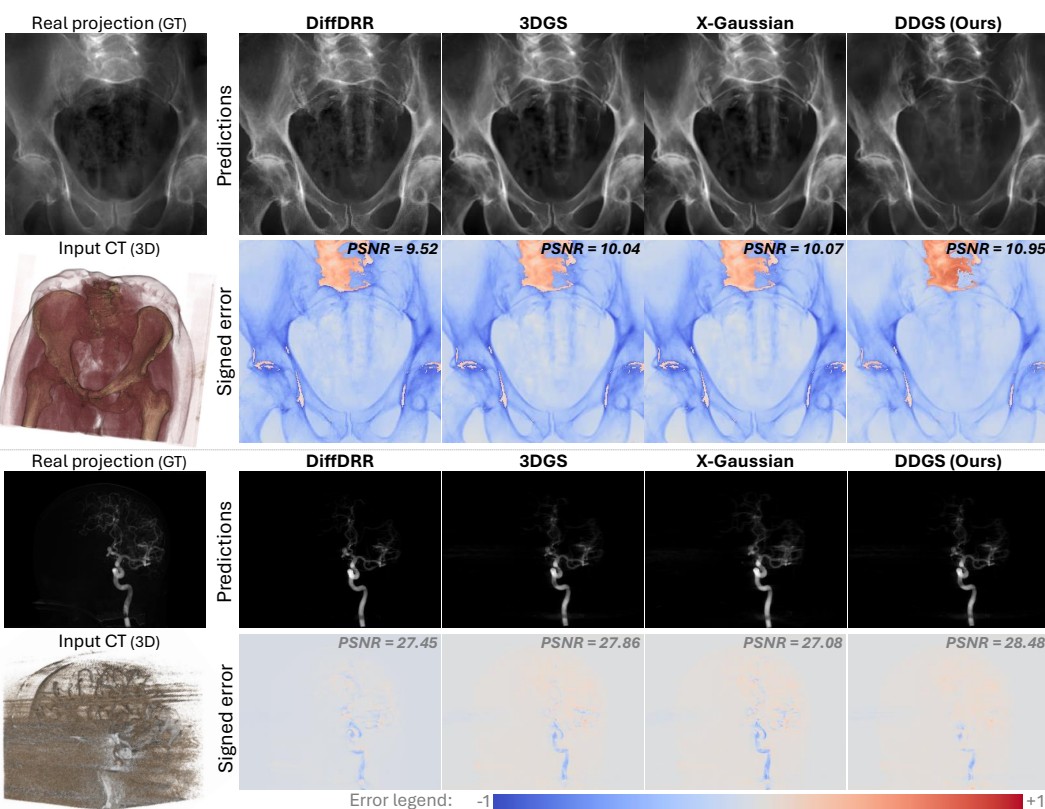

Figure 4: Qualitative comparison of DRRs and real scans from DeepFluoro and Ljubljana datasets.

focus more on bones or anatomical structures, while the direction-independent 3D Gaussians focus more on the background.

Table 4 shows the effectiveness of our proposed initialization scheme (radiodensity-aware) compared to other initialization methods. Our scheme outperforms both the uniformly random sampling used in 3DGS [17] and the even sampling method used in X-Gaussian [7].

At last, we conduct experiments on the impact of feature dimension $k$, as shown in Table 5. Our results indicate that increasing the feature dimension generally improves overall performance. However, the improvement becomes marginal once the feature dimension exceeds $k = 16$.

Table 4: Impact of initialization strategies on image quality during training, *i.e.*, measured on CTPelvic1K scans w.r.t. training iterations.

| Iteration | Strategy | 001 | 002 | 003 | 004 | 005 | 006 | 007 | 008 | 009 | 010 | avg |
|---|---|---|---|---|---|---|---|---|---|---|---|---|
| 500 | RADS (**ours**) | **25.86** | **27.72** | **25.24** | **28.52** | **30.52** | **28.31** | **29.23** | **27.92** | **28.95** | **29.20** | **28.15** |
| | Random | 25.29 | 26.73 | 24.66 | 27.40 | 28.46 | 26.35 | 27.23 | 26.43 | 27.34 | 28.53 | 26.84 |
| | Even | 24.96 | 26.64 | 24.40 | 26.60 | 27.54 | 26.63 | 25.06 | 26.01 | 26.15 | 26.92 | 26.09 |
| 7,000 | RADS (**ours**) | **35.02** | **37.06** | **36.47** | **36.49** | **36.79** | **34.52** | **35.56** | **36.03** | **36.44** | **35.55** | **35.99** |
| | Random | 34.39 | 35.78 | 34.18 | 35.83 | 36.37 | 34.36 | 35.20 | 35.74 | 36.30 | 35.26 | 35.34 |
| | Even | 34.22 | 36.07 | 33.65 | 36.31 | 36.64 | 34.13 | 34.97 | 35.85 | 35.61 | 35.33 | 35.28 |
| 30,000 | RADS (**ours**) | **37.88** | 38.43 | **38.28** | **40.35** | **40.36** | 38.26 | **38.87** | 39.81 | **39.87** | 38.73 | **39.08** |
| | Random | 37.17 | 38.36 | 38.15 | 40.09 | 39.97 | **38.36** | 38.69 | 39.37 | 39.85 | 38.84 | 38.89 |
| | Even | 37.33 | **38.51** | 37.83 | 39.58 | 39.12 | 38.27 | 38.87 | **39.88** | 39.46 | **38.95** | 38.78 |

Table 5: Impact of $k$ dimensionality on image quality, measured on 2 CTPelvic1K scans (001, 002).

| | $k = 1$ | | $k = 4$ | | $k = 8$ | | $k = 16$ | | $k = 32$ | |
|---|---|---|---|---|---|---|---|---|---|---|
| | PSNR | SSIM | PSNR | SSIM | PSNR | SSIM | PSNR | SSIM | PSNR | SSIM |
| 001 | 34.60 | 0.965 | 36.81 | 0.979 | 38.04 | 0.984 | **38.41** | 0.986 | 38.27 | **0.987** |
| 002 | 33.66 | 0.964 | 37.08 | 0.982 | 38.30 | 0.987 | 38.76 | 0.989 | **38.89** | **0.989** |

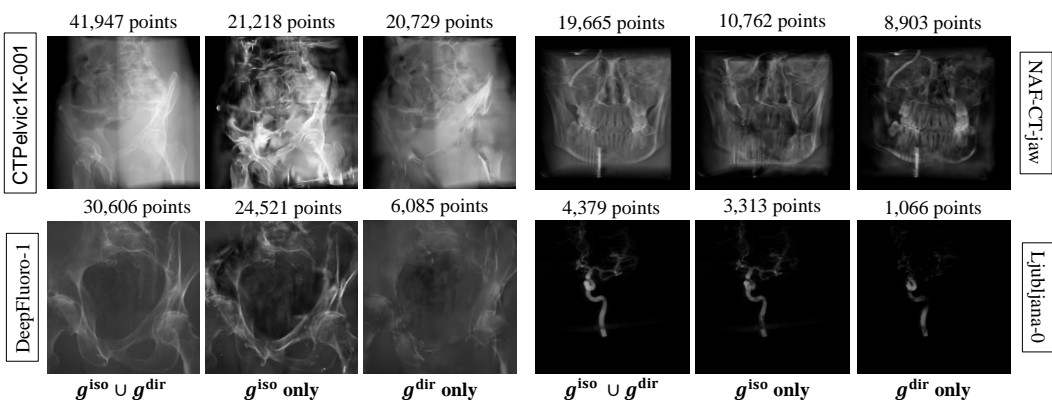

Figure 5: Visualization of the isotropic and anisotropic X-ray contributions to the rendered DRRs.

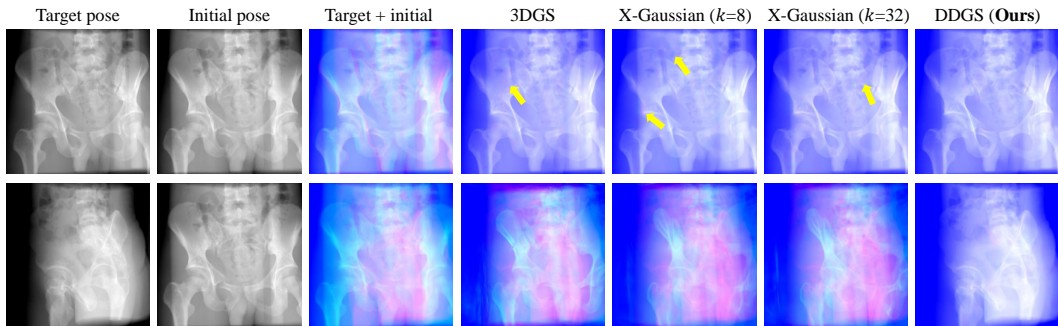

Figure 6: Illustration of the pose-registration convergence using various differentiable DRR renderers.

Table 6: Accuracy of image registration on the CTPelvic1K [19] dataset.

| | 3DGS [17] | | | X-Gaussian [7] ($k$=8) | | | X-Gaussian [7] ($k$=32) | | | DDGS (**Ours**) | | |
|---|---|---|---|---|---|---|---|---|---|---|---|---|
| | 001 | 002 | 003 | 001 | 002 | 003 | 001 | 002 | 003 | 001 | 002 | 003 |
| Rot. error (deg) | 0.134 | **0.062** | 0.229 | 0.326 | 0.163 | 0.312 | 0.097 | 0.070 | 0.266 | **0.067** | 0.067 | **0.152** |
| Trans. error (mm) | 1.788 | 1.163 | 3.108 | 4.037 | 2.144 | 3.781 | 1.378 | 1.288 | 3.483 | **1.285** | **1.092** | **2.180** |

Table 7: Registration evaluation w.r.t. anatomical landmark accuracy and time efficiency, on Ljubljana.

| Method | DRR render time (ms) ↓ | | | total optimization time (s) ↓ | | | TRE (mm) ↓ | | |
|---|---|---|---|---|---|---|---|---|---|
| | mean | med | std | mean | med | std | mean | med | std |
| DiffDRR [11] | 30.88 | 31.51 | 3.25 | 431.98 | 316.54 | 261.70 | 2.14 | 0.72 | 5.40 |
| DDGS | **7.46** | **6.94** | **2.67** | **121.45** | **39.93** | **117.67** | **0.80** | **0.65** | **0.51** |

### 4.3 Application to Downstream Task (2D/3D CT Image Registration)

We validate our solution in terms of applicability to downstream tasks, considering here the registration of intraoperative 2D X-ray images to preoperative 3D CT volumes; a task crucial to a variety of clinical applications.

We choose the first five testing images on the 001, 002, and 003 CT data of CTPelvic1K [19] for image registration. Considering this task as an optimization-based problem that can be solved via inverse graphics (*i.e.*, rendering synthetic images based on predicted poses and comparing to the target image, then backpropagating the difference to improve the pose prediction), we adopt the framework from iComMA [32], replacing their image rendered by DDGS. We use Adam optimizer with a learning rate of 0.05. We use the isocenter pose as the initialization pose. As shown in Table 6 and Figure 6, our DDGS achieves the lowest rotation and translation errors.

Finally, we also consider the experimental protocol proposed in DiffPose [12], where the authors rely on a pretrained pose-regression CNN to get a first rough estimation of the scanner pose, before refining said pose via a gradient-descent based optimization, leveraging their differentiable DRR renderer [11]. We adopt their pose-initialization network and testbed for the Ljubljana dataset, replacing their DiffDRR analytical renderer [11] by our DDGS, and compare to them in terms of runtime performance and landmark-registration accuracy. The results, reported in Table 7, highlight the convergence boost brought by our lighter solution.

## 5 Conclusion

We proposed DDGS, a novel 3DGS-based representation for efficient and realistic DRR generation from CT volumes. We extended traditional 3DGS by considering light directionality, crucial for accurate X-ray image modeling. By decomposing radiosity into non-directional and direction-dependent components, DDGS captures isotropic interactions and anisotropic effects. Unlike generic view-dependent spherical-harmonics decomposition, DDGS implicitly learns scattering effects, substantially improving the speed and accuracy of downstream clinical applications such as intraoperative 2D/3D registration.

**Limitations.** It should be noted that, similar to previous methods, our model expects an offline DRR renderer to produce realistic ground truth to train DDGS, which can be a limitation for some scenarios (*e.g.*, when such renderer or certain parameters required for realistic rendering are not available). Our direction-dependent function relating to anisotropic X-ray effects may also fail to approximate complex multi-bounce light scenarios. Recent developments in $N$-dimensional Gaussian splatting [8] show promising potential for modeling such cases. We should also note that the quality of DRRs is bound to the quality of the input CT scans. This issue affects both DRR methods and Monte-Carlo simulations (as the latter relies on the segmentation of CT scans into materials-specific regions – if a 3D scan is noisy, so will the material segmentation and simulation results). When our method uses simulators to get target images, it will face similar noise issues, though, our experiments show that DDGS barely introduces additional noise. Compensating for CT noise is an interesting, under-explored research direction that could benefit all DRR methods; but, we believe it is beyond the scope of our study.

**Societal Impact.** The above limitations may restrict the adoption of DDGS to specific use cases, and our method should undergo a clinical evaluation of the generated images in terms of anatomical accuracy. Nonetheless, we believe that our proposed method can positively impact both the computer vision and medical communities by providing a more efficient and versatile DRR tool.

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

# Supplementary Material

In this supplementary material, we provide further methodological context and showcase additional quantitative and qualitative results to highlight further the contributions claimed in the paper.

## A  Methodological Insight

### A.1  Transmittance in Natural vs. X-Ray Imaging

The exponential transmittance model $T(t)$ used in NeRF and Gaussian-splatting (GS) solutions for natural scenes—to describe the light attenuation as it travels through a medium from point $r(t_0)$ to $r(t)$—is based on Beer-Lambert law [16, 23, 37, 33]:

$$T(t) = \exp(-\int_{t_0}^{t} a(r(s))\delta s), \tag{4}$$

where $r(t) = o + td$ is a 3D point at step $t$ on a ray of origin $o$ and direction $d$, and $a(p)$ is the medium's absorption/extinction (linked to its density) at point $p$. Assuming a piece-wise homogeneous medium (with $a_i$ the absorption of homogeneous segment $l$ of length $\delta_l$), this transmittance can be rewritten:

$$T(t) = \exp(-\sum_{l=1}^{l_t} a_l \delta_l) = \prod_{l=1}^{l_t} (1 - \sigma_l), \tag{5}$$

as in our Equation 3.1; with $\sigma_l = 1 - \exp(-a_l \delta_n)$ and $l_t$ the end segment containing $t$. See [33] for proof.

X-ray attenuation from photoelectric effect (PE) follows the same model: at each step through the medium (*e.g.*, voxel), the probability of photon extinction (*i.e.*, averaged attenuation) is proportional to the atomic density. *E.g.*, if a ray travels through only 2 voxels, and if it is attenuated by $a_1 = 50\%$ at step 1 then $a_2 = 50\%$ again at step 2, then its total attenuation is 75% (*c.f.* $(100 - (100 - 50)^2)\%$), in accordance to Beer-Lambert.

Hence, Equation 3.1 (3DGS rendering) could be directly applied to DDR synthesis (assuming standard *neglog* scaling of absorption values into pixel ones [12, 35]), if we were to adopt a simplified model of X-ray imaging which only considers isotropic absorption (*i.e.*, following the Beer-Lambert law) and optionally omit the term $c_j \in \mathbb{R}^K$ (view-dependent radiance). This is the simplified model proposed in prior work [7].

In our paper, we propose to go beyond the isotropic simplification in X-Gaussian [7] and further account for anisotropic scattering of X-rays. This is why we preserve $c_j$ and decompose it into two terms: an isotropic term $c^{\text{iso}}$ and direction-dependent non-linear term $c^{\text{dir}}$ (Equation 3.2). Note that we also fix the dimension $K$ of these variables (*i.e.*, the number of output channels) to 1 (monochromatic DRR) instead of three in natural 3DGS (RGB).

### A.2  Joint Gaussian Rasterization

As explained in Section 3, we define two distinct functions for the absorption contribution of the two Gaussian sets: one function is isotropic to approximate average radio-absorption ($c_i^{\text{iso}}$), and one function is direction-dependent to approximate the contribution of anisotropic Compton scattering to the image ($c_j^{\text{dir}}$), *c.f.* Equation 3.2.

These two distinct Gaussian sets could be rasterized separately, as done in Figure 5 as qualitative ablation. However, the two sets *need to be rendered together to ensure correct simulation*. Otherwise, Gaussians from one set occluded by others belonging to the other set could incorrectly contribute to the ray absorption. We refer the readers to Figure S1 for an explanatory diagram. There, point $g_2^{\text{dir}}$ should not contribute to the final pixel value, as it is occluded by $g_2^{\text{iso}}$ (*i.e.*, it absorbs all the remaining energy before the ray can reach $g_2^{\text{dir}}$). This could not be properly simulated if each Gaussian set is rasterized separately.

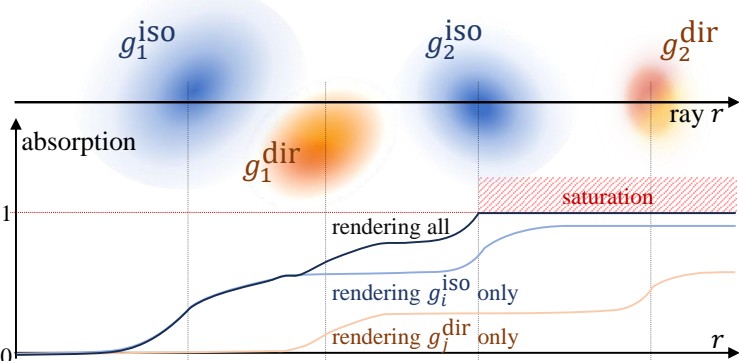

Figure S1: Importance of joint rasterization of Gaussian sets (simplified diagram).

By having 2 sets of Gaussians with distinct yet complementary contribution functions (isotropic versus direction-dependent functions), our solution can better model DRR imaging. During optimization, each Gaussian set will approximate its respective imaging effect, conditioned by its respective function.

We randomly split $\mathbf{P}^{mc}$ (3D points sampled based on the CT scan distribution) into 2 sets, each subset contributing to the initialization positions of $g_i^{iso}$ and $g_j^{dir}$ (half the 3D points in $\mathbf{P}^{mc}$ will serve as initialization for $g_i^{iso}$, along with the entire $\mathbf{P}^{mc}$ set; and the other half of $\mathbf{P}^{mc}$ will serve as initialization for $g_j^{dir}$). After initialization, during 3DGS-based model optimization, the points in each set of Gaussians can move/split/drop independently (along with the optimization of their respective covariance/opacity/feature values). Since the optimization of each set (isotropic and anisotropic) is conditioned on their respective differentiable absorption function, they will acquire distinct properties.

## B  Additional Experiments

### B.1  Additional Comparison and Ablation Results

To further support the results in Table 2 and Table 3, we evaluate our method on all the 10 selected data from the CTPelvic1K dataset, as shown in Table S1.

Table S1: Novel-view synthesis evaluation on CTPelvic1K data.

|  | DDGS (ours) ($L=1, k=8$) | | 3DGS ($L=1$) | | X-Gaussian ($k=8$) | | DDGS (ours) ($L=3, k=8$) | | 3DGS ($L=3$) | | X-Gaussian ($k=32$) | | direct-entangled *c.f.* Ablation Study | |
|---|---|---|---|---|---|---|---|---|---|---|---|---|---|---|
|  | PNSR | SSIM | PNSR | SSIM | PNSR | SSIM | PNSR | SSIM | PNSR | SSIM | PNSR | SSIM | PNSR | SSIM |
| 001 | **38.04** | **.984** | 35.36 | .970 | 33.39 | .947 | **38.15** | **.985** | 37.12 | .980 | 36.83 | .979 | 36.17 | .977 |
| 002 | **38.30** | **.987** | 37.25 | .982 | 36.04 | .978 | **38.97** | **.989** | 38.75 | .988 | 37.89 | .986 | 37.32 | .984 |
| 003 | **38.21** | **.983** | 35.81 | .974 | 33.99 | .965 | **39.42** | **.986** | 39.09 | .984 | 35.84 | .976 | 35.32 | .975 |
| 004 | **40.54** | **.987** | 38.43 | .980 | 37.18 | .974 | **40.74** | **.987** | 40.69 | .986 | 39.06 | .984 | 39.70 | .985 |
| 005 | **39.88** | **.987** | 39.44 | .985 | 37.50 | .978 | 40.02 | .987 | **40.68** | **.989** | 39.24 | .985 | 39.36 | .986 |
| 006 | **38.12** | **.986** | 36.69 | .982 | 35.03 | .977 | **38.79** | .986 | 38.65 | **.987** | 37.14 | .983 | 36.89 | .983 |
| 007 | **38.91** | **.987** | 36.88 | .978 | 35.51 | .970 | **39.37** | **.988** | 38.58 | .984 | 38.10 | .987 | 37.72 | .985 |
| 008 | **39.83** | **.982** | 37.85 | .979 | 36.70 | .973 | **40.19** | **.986** | 39.70 | .986 | 38.57 | .982 | 38.53 | .982 |
| 009 | **39.67** | **.987** | 37.94 | .980 | 37.60 | .979 | **40.35** | **.988** | 39.16 | .983 | 38.99 | .987 | 38.74 | .985 |
| 010 | **38.60** | **.987** | 37.50 | .982 | 36.30 | .978 | **39.18** | **.988** | 39.13 | .986 | 37.89 | .986 | 37.20 | .985 |
| avg | **39.01** | **.986** | 37.32 | .979 | 35.92 | .972 | **39.52** | **.987** | 39.16 | .985 | 37.96 | .984 | 37.70 | .983 |

### B.2  Comparison to Real X-ray Projections

The main purpose of DRRs is to provide quick visualization to clinicians (with the key metrics being speed and visibility), as well as to integrate larger imaging applications (with priority given again to speed and feature-level similarity w.r.t. real data). Therefore, prior DRR papers (traditional [11, 30, 31] or GS-based [7, 25]) mostly evaluate on downstream tasks (*e.g.*, pose registration). A few evaluate the image quality compared to other DRR tools (DiffDRR [11] compares to Siddon's

Table S2: Quantitative comparison of analytical DRRs to real images from Ljubljana dataset.

| Methods | SSIM | | | PSNR | | |
|---|---|---|---|---|---|---|
| | mean$^\uparrow$ | med$^\uparrow$ | std$^\downarrow$ | mean$^\uparrow$ | med$^\uparrow$ | std$^\downarrow$ |
| DiffDRR [11] | .404 | .440 | **.179** | 24.16 | 24.35 | 2.82 |
| 3DGS [17] | .409 | .518 | .199 | 22.57 | 23.44 | **2.78** |
| X-Gaussian [7] | .409 | .521 | .198 | 22.58 | 23.42 | **2.78** |
| DDGS (ours) | **.459** | **.558** | .204 | **25.61** | **27.35** | 3.48 |

and Plastimatch [30]; [7] to other NVS baselines). We found that only DiffPose (based on DiffDRR) [12] provides a qualitative comparison to some real scans.

Quantitative comparison to real data is challenging. Not all imaging parameters are usually provided or accurate enough to create matching DRRs (w.r.t. intensity, CT pose, *etc*.). It is especially hard to render DRRs aligned with real scans with existing Monte-Carlo (MC) simulation tools [2–4, 15]; which is likely why *no DRR paper has performed such evaluation*. Their interfaces and custom coordinate conventions are not compatible with pose annotations in public CT/X-ray datasets, and their documentation/support is lacking. Bridging the convention gap between MC and analytical communities would greatly benefit our domain and could be the focus of our next effort.

Nevertheless, we do provide an indirect comparison to real projections. *E.g*., the registration experiments (Table 6) implicitly provide insight into the similarity between real and synthetic data, as pose optimization is done by comparing DRRs to real target scans.

In this supplementary, we share a more direct comparison. We use our DDGS-based registration method to refine GT poses in Ljubljana data (*c.f*. aforementioned GT inaccuracies), then use the refined poses to render and compare DRRs to real scans. Results can be found in Table S2 and Figure 4. *E.g*., when zooming into the images of the latter figure, one can observe that DiffDRR suffers from pixelization, unlike GS methods.

