# OpenReview forum: "DDGS-CT: Direction-Disentangled Gaussian Splatting for Realistic Volume Rendering"
_NeurIPS.cc/2024/Conference — NeurIPS 2024 poster_

### Official Review · Reviewer_g5Vp · 2024-07-09

**Soundness:** 2
**Presentation:** 2
**Contribution:** 2
**Rating:** 4
**Confidence:** 4

**Summary:**

1. The proposed direction-disentangled 3DGS (DDGS) method decomposes the radiosity contribution into isotropic and direction-dependent components, able to approximate complex anisotropic interactions without complex runtime simulations. specifically, it modeling isotropic and anisotropic contributions via distinct 3D Gaussians.

2. The paper clearly mentioned that 3DGS-for-DRR solutions [30, 11] do not account for noise-inducing photon interactions (e.g., scattering) when applying the analytical methods [29, 5] to render their training data.

**Strengths:**

1. This paper has a clear definition of the problem to be solved and a good visualization.
2. Experiments combining 2D/3D CT Image registration were also presented.

**Weaknesses:**

1. Although Gaussian Splatting is widely used in natural color scenes, this paper does not provide a detailed mathematical description of the "migrated version" of Gaussian Splatting in combination with the physical scene of CT.  Instead, it directly use the mathematical description of Gaussian Splatting in natural color scenes (Equation (1)). In CT imaging, the contribution of a voxel to the final projection does not suffer from attenuation due to occlusion by the following voxels, but is instead Beer-Lambert law. Therefore, equation (1) no longer holds. This makes it difficult to highlight the motivation of this paper in terms of algorithm details.

2.  Theoretical contributions appear to be minor. (Formula (3) is the theoretical contributions of this paper). This makes this paper more suitable for delivery to places that are more concerned with simulation applications.

3. The paper needs to compare with Physics-based Monte-Carlo simulations [2, 15, 3 , 4 ]. Although these methods are time-consuming, they guarantee the quality of reprojection. The paper needs to clearly point out the difference in image quality with Physics-based Monte-Carlo simulations. The ground truth(GT) needs to select the real projection obtained by the CT machine. So the comparison table will be：
______________________________________________________________________________________
                         |    Physics-based Monte-Carlo simulations     |     3DGS   |  X-Gaussian  |   DDGS
_____________________________________________________________________________________
PSNR with GT |
______________________________________________________________________________________
SSIM  with GT |
______________________________________________________________________________________

**Questions:**

1. what is the means of high-dimensional residual contribution？Is there a formal description? This concept is not explained in the full text.

2. Before use the method, it need first get the 3D CT volume. The reconstruction is based on the projection obtained by the CT machine. However, there are errors and noise in the 3D CT volume (from the reconstruction of the projection noise) and also from the reconstruction errors caused by the polychromatic X-rays in medical CT machine. The paper is equivalent to reprojecting the 3D CT volume using gaussian splatting. How do the work ensure the reprojection process does not amplify this error from the 3D CT volume?

**Limitations:**

1. If this method is used in clinical practice, what problems will it cause without considering the time cost? For example, the appearance of artifacts, because once artifacts appear, it will seriously interfere with the doctor's clinical operation.

---

> ### Author Rebuttal · Authors · 2024-08-06
>
> ### Beer-Lambert law:
> We would like to kindly correct the reviewer. The exponential transmittance model $T(t)$ used in NeRF/GS, to describe the light attenuation as it travels through a medium from point $r(t_0)$ to $r(t)$, is **also based on Beer-Lambert law** [17, r1-4]:
>
> $T(t)=\exp(-\int_{t_0}^t a(r(s))\delta s)$,
>
> where $r(t)=o+td$ is a 3D point at step $t$ on a ray of origin $o$ and direction $d$, and $a(p)$ is the medium's absorption/extinction (linked to its density) at point $p$. Assuming a piecewise homogeneous medium (with $a_i$ the absorption of homogeneous segment $l$ of length $\delta_l$), this transmittance can be rewritten:
>
> $T(t)=\exp(-\sum_{l=1}^{l_t} a_l\delta_l)=\prod_{l=1}^{l_t}{(1-\sigma_l)}$,
>
> as in our Eq. 1; with $\sigma_l=1-\exp(-a_l\delta_n)$ and $l_t$ the end segment containing $t$. See [r4] for proof.
>
> X-ray attenuation from photoelectric effect (PE) follows the same model: at each step through the medium (e.g., voxel), the probability of photon extinction (i.e., averaged attenuation) is proportional to the atomic density. E.g., if a ray travels through only 2 voxels, and if it is attenuated by $a_1$=50% at step 1 then $a_2$=50% again at step 2, then its total attenuation is 75% (c.f. $(100-(100-50)^2)$%), in accordance to Beer-Lambert.
>
> Hence, Eq. 1 (3DGS rendering) could be directly applied to DDR synthesis (assuming standard _neglog_ scaling of absorption values into pixel ones [12, 33]), if we were to adopt a simplified model of X-ray imaging which only considers isotropic absorption (i.e., following the Beer-Lambert law), though the term $c_j\in\mathbb{R}^{K}$ (view-dependent radiance) which could be omitted. This is the simplified model proposed in prior work [7], as well as concurrent work [r5] (published online after our submission).
> In our paper, we propose to go beyond their isotropic simplification and further account for anisotropic scattering of X-rays. This is why we preserve $c_j$ and decompose it into two terms: an isotropic term $c^{\text{iso}}$ and direction-dependent non-linear term $c^{\text{dir}}$ (Eq. 3). Note, we also fix the dimension $K$ of these variables (= number of output channels) to 1 (monochromatic DRR) instead of 3 in usual 3DGS (RGB). We will clarify this in the paper.
>
> ### Contributions w.r.t. 3DGS:
> We would argue that while Eq. 3 is compact, it is not straightforward; nor it covers all our contributions (e.g., radiodensity-aware initialization). It may be easily overseen as it is formalized via subscripts ($i$ vs. $j$) but a key contribution is the decoupling of the isotropic and view-dependent terms, modeled by distinct Gaussians ($g^{\text{iso}}_i$ and $g^{\text{dir}}_j$) c.f. L163-164 and 171-177. This differs from prior 3DGS and DRR work [7, 24], using 1 set of Gaussians. Our dual-set idea may seem _unintuitive_ but demonstrates better compactness and accuracy. We believe **this insight benefits the community**.
>
> In comparison, X-Gaussian [7] (ECCV 2024) adapts 3DGS to DRR rendering by: (a) simplifying view-dependent radiance into an isotropic term (similar to our $c^{\text{iso}}$); and (b) replacing SfM initialization with uniform 3D space sampling (ignoring CT content). **Our contributions are comparatively significant**: (a) a 2nd anisotropic term $c^{\text{dir}}$ with its own Gaussians $g^{\text{dir}}$ (a more accurate physics adaptation of 3DGS to X-rays); and (b) an advanced initialization strategy considering CT data distribution; overall resulting in a **+1.04db PSNR increase** and a **10% decrease in needed Gaussians** compared to [7] over CTPelvic1K (Tab. 1).
>
> ### MC/real comparison:
> Please refer to our global response. We cannot compare MC and analytical solutions against real images but:
> - Tab. 1 already provides a relevant comparison between DRR tools (as candidates) and MC simulation (TIGRE [5], as GT). The high metrics for DDGS show that it approximates MC simulation well.
> - Similarly, the registration experiments (e.g., Tab. 7) implicitly provide insight into the similarity between real and synthetic data, as pose optimization is done by comparing DRRs to real target scans.
> - For a more direct comparison, we used our DDGS-based registration method to refine GT poses in Ljubljana data (c.f. aforementioned GT inaccuracies), then used the refined poses to render and **compare DRRs to real scans**. Results can be found in Tab. R1 and Fig. R1 (rebuttal PDF). E.g., when zooming, we notice that DiffDRR suffers from pixelization, unlike GS methods. Refined GT data will be made public.
>
> ### High-dim. residual contribution:
> Thank you, we will clarify. X-ray scattering is a complex phenomenon causing imaging noise (residual impact). The noise distribution is influenced by so many parameters (source direction and energy, medium properties, bounces, etc.), only a costly high-dimensional function could accurately model it (see Sec. 2.1 of [8] for details). Therefore, we only claim to approximate the residual impact of scattering.
>
> ### CT noise:
> We agree that the quality of DRRs is bound to the quality of the input CT scans. This issue affects all DRR and MC methods. E.g., MC-based tools segment CT scans into materials and use the segmented data as simulation input. If a 3D scan is noisy, so will the material segmentation and simulation results (see discussion in [33]). When our method uses simulators to get target images, it will face similar noise issues. However, our experiments show that DDGS barely introduces additional noise (highest PSNR in Tabs. 1-2). Compensating for CT noise is an interesting, under-explored research direction that could benefit all DRR methods; but, we believe, beyond the scope of our study.
>
> ### Clinical usage:
> When clinicians use DRRs for quick visualization, they know the tools' limitations. E.g., DRRs are typically not used for diagnosis. We agree with the reviewer that image quality is still important, hence our experiments showing DDGS accuracy over similar DRR works.

---

> > ### Comment · Reviewer_g5Vp · 2024-08-10
> >
> > Thanks, I think the author's answer solved some of my doubts, but I decided to keep my score.

---

> ### Author Response · Authors · 2024-08-10
>
> We appreciate `g5Vp`'s feedback and are glad to hear that "_the author's answer solved some of my doubts._" Given that some of the previously mentioned weaknesses have been addressed, may we respectfully ask `g5Vp` to kindly elaborate on their decision to nevertheless opt for _borderline reject_? We are eager to address any remaining concerns and to further improve our submission based on the feedback.
>
> In our view, we have responded to the key concerns raised by `g5Vp`, specifically:
>
> - **Weakness #1 = theoretical misunderstanding by `g5Vp`**. We hope that we have clarified the application of the Beer-Lambert Law to correct the reviewer's understanding.
>
> - **Weakness #2 = significant contributions compared to SOTA**. We acknowledge that the novelty and impact of a paper are subjective, so we have highlighted the theoretical and empirical significance of our contributions relative to the SOTA  [7, 11, 12].
>
> - **Weakness #3 = additional results provided**. We exceeded the standard analytical DRR evaluation [5, 7, 11, 12, 33] by comparing our solution to real projections, as requested by the reviewer.
>
> We have also thoroughly addressed the questions of other reviewers, particularly those raised by `ZGY5` regarding the SOTA status of [7, 11, 12] and the paper’s organization. We note that overall, the reviewers' concerns/questions do not have major overlaps, as opposed to their positive feedback ("novelty" [`Zg8U`, `YFhr`, `ZGY5`], "substantial improvement on target tasks" [`all`], "clear motivation/methodology" [`all`], etc.).
>
> Finally, we would like to emphasize that we submitted this paper to **NeurIPS's "_Machine learning for healthcare_" primary area**. We believe our solution is **well-suited to this application area**, given its ML contributions and focus on clinical tasks. Thus, we respectfully disagree with the reviewer's suggestion that our submission might be better suited to a different venue.
>
> We kindly ask `g5Vp` to consider the arguments above; and if `g5Vp` still believes that our submission is not appropriate for NeurIPS (_Machine learning for healthcare_), we would be thankful for an explanation.
>
> Sincerely,
>
> The Authors

---

### Official Review · Reviewer_ZGY5 · 2024-07-10

**Soundness:** 2
**Presentation:** 2
**Contribution:** 2
**Rating:** 3
**Confidence:** 4

**Summary:**

Digitally reconstructed radiographs (DRRs) are simulated 2D X-ray images generated from 3D CT volumes, widely used in preoperative settings but limited in intraoperative applications due to computational bottlenecks. In this paper, the author proposes an approach that combines realistic physics-inspired X-ray simulation with efficient, differentiable DRR generation using 3D Gaussian splatting (3DGS). This method excels in both image accuracy and inference speed, demonstrating its potential for intraoperative applications and inverse problems like pose registration.

**Strengths:**

Digitally reconstructed radiographs (DRRs) are simulated 2D X-ray images generated from 3D CT volumes, widely used in preoperative settings but limited in intraoperative applications due to computational bottlenecks. In this paper, the author proposes an approach that combines realistic physics-inspired X-ray simulation with efficient, differentiable DRR generation using 3D Gaussian splatting (3DGS). This method excels in both image accuracy and inference speed, demonstrating its potential for intraoperative applications and inverse problems like pose registration.

**Weaknesses:**

1. The methods used for comparison are limited and do not reflect the current state of the field. The state-of-the-art methods claimed by the author are not widely accepted as state-of-the-art by researchers, and some have not been peer-reviewed, like reference [7,11]. Therefore, I believe the experimental results lack persuasiveness.
2. The structure of the paper is not well organized. Despite the author clearly stating the motivation and contributions, the intentions of the author remain unclear. I have a rough idea of what the related work section is trying to convey, but its organization is also poor. It is recommended to include the 'Application' from section 3.1 within the related work.
3. In the ablation experiments, only a portion of the data was tested. Results from testing the entire dataset would be more convincing.

**Questions:**

1. The methods section lacks explanations for some symbols. Please review and provide explanations where necessary. Additionally, are g_i^{iso} and c_i^{iso}, as well as g_i^{dir} and c_i^{dir}, referring to the same concepts? Please review carefully. For the methods section, it is recommended to include an overview diagram for clarification.
2. Add more true sota methods for comparison.

**Limitations:**

1. The experiments are insufficient. The comparison method lacks representativeness and fails to demonstrate the true level of the field.
2. The method description is confusing and the contributions are not clear.

---

> ### Author Rebuttal · Authors · 2024-08-06
>
> ### SOTA status of X-Gaussian and DiffDRR/DiffPose:
> We respectfully disagree with the reviewer w.r.t. where the SOTA stands. Since we started writing this paper, DiffPose [12] (an extension of DiffDRR [11] by the same authors) has been presented at **CVPR 2024 (_oral_)**, and X-Gaussian [7] has been approved at **ECCV 2024**. We believe that this demonstrates that our analysis of the state of the field is correct and that our **comparison to these methods is essential**. Note that, at a fundamental level, we do agree that the current trend in machine learning to consider preprints as legitimate work can be counterproductive to science's steady progress. In this case, however, we made a decision, now validated by peers, based on the quality of these works [7, 11, 12] and our knowledge of the field.
>
> The application of novel inverse graphics techniques (e.g., 3DGS) to DRR rendering is a fast-moving field, but we did our best to reference the relevant methods. We especially consider X-Gaussian [7] and DiffDRR/DiffPose [11, 12] to be the most important for comparison, as they **each represents the state-of-the-art in their respective type of analytical DRR rendering** (X-Gaussian for 3DGS-based DRR rendering, and DiffPose as a differentiable extension of Siddon's voxel-based algorithm [30] enabling optimization-based tasks), directly relevant to our work.
>
> ### Manuscript organization:
> We thank the reviewer for finding our motivation and contributions clear, and for the suggestion to shuffle a bit the Related Work and Preliminaries sections. We agree that our manuscript may benefit from moving some application-related sentences to the former section.
>
> Regarding the suggestion to express our _intentions_ as clearly as our _motivation_:
> our key goal is to define a DRR rendering solution that (a) is fast and differentiable in order to enable intraoperative tasks (e.g., CT/X-ray registration); but also that (b) matches more closely the complex X-ray imaging process without sacrificing performance. Our intuition is that a better trade-off between realism and computational performance (compared to recent 3DGS-based solutions for DRRs, which ignore anisotropic properties) could further help downstream tasks, as demonstrated in the paper (accurate MC simulation approximation and pose estimation).
>
> ### Additional ablation results:
> We understand that performing some of the ablation studies on data subsets is not ideal. We believed that the selected instances were statistically representative enough and wanted to preserve space and computing power for the other quantitative results. It is not ideal, but also common with rich 3D datasets to perform ablation studies on a subset [8, 17, 31, 35, 36, etc.].
>
> We **complete our ablation studies**, with results shared **in Tables R2-R3** (in the rebuttal PDF). More will be added to the paper and supplementary material.
>
> ### Symbol explanations:
> Could the reviewer specify which unexplained symbols they are referring to, please? Note that we went through our manuscript and found 1 typo, L208: we meant to write "loss weight $\lambda$" rather than "loss weight $\delta$" (referring to the symbol in Equation 2). We thank the reviewer for helping us correct this.
>
> Regarding $g_i^{\text{iso}}$ and $c_i^{\text{iso}}$ (as well as $g_j^{\text{dir}}$ and $c_j^{\text{dir}}$), those are different concepts (L163-165). We used the notation $g$ to represent Gaussians (each defined by its 3D mean position, covariance, opacity, and color/intensity features c.f. L113). So $g_i^{\text{iso}}$ and $g_j^{\text{dir}}$ each represents a distinct set of Gaussians, with different 3D positions, covariances, opacities, and, more importantly, different feature vectors (resp. isotropic $\mathbf{f}_i^{\text{iso}}$ and direction-dependent $\mathbf{f}_j^{\text{dir}}$).
>
> In contrast, we used the symbol $c$ in the paper to represent the color/absorption contribution of each Gaussian to the image (c.f. Equation 1). Therefore, $c_i^{\text{iso}}$ represents the contribution of the Gaussian $g_i^{\text{iso}}$, calculated according to Equation 3 (i.e., multiplying its feature vector $\mathbf{f}_i^{\text{iso}}$ to the global isotropic basis vector and then applying a sigmoid activation), and similarly for $c_j^{\text{dir}}$ being the contribution of $c_j^{\text{dir}}$ to the pixel, after applying our proposed anisotropic model (c.f. Equation 3).
>
> We would gladly take into account any further suggestions to make our manuscript accessible to a larger audience. E.g., we will improve our pipeline figure (Fig. 1) by adding the above-mentioned notations within, in order to help the readers understand their application.

---

> > ### Comment · Reviewer_ZGY5 · 2024-08-12
> >
> > After reading the author's response and the other reviews, I would like to keep my initial score.

---

> > > ### Author Response · Authors · 2024-08-12
> > > **Response to Reviewer ZGY5**
> > >
> > > Dear `ZGY5`,
> > >
> > > Thank you for your message.
> > >
> > > We believe that we have addressed your comments, as well as those from other reviewers, in our responses. However, we are concerned about the blank “rejection” decision and would appreciate if more **detailed** and **concrete** reasons for this outcome could be provided. This would greatly help us understand the reviewer's perspective and further improve our work.
> > >
> > > Sincerely,
> > >
> > > The authors

---

### Official Review · Reviewer_YFhr · 2024-07-10

**Soundness:** 3
**Presentation:** 3
**Contribution:** 3
**Rating:** 6
**Confidence:** 3

**Summary:**

This paper proposes DDGS, a Gaussian splatting (GS) based method for rendering realistic 2D X-ray images from 3D CT volumes. Taking advantage of GS, the proposed method operates in real-time. Moreover, the DDGS model employs isotropic Gaussians and anisotropic, direction-dependent Gaussians to model complex X-ray physical interactions, such as Compton scattering. The proposed method is well-motivated and technically sound. The effectiveness of the DDGS model is confirmed on multiple datasets.

**Strengths:**

**Motivation**

Conventional Monte-Carlo methods can produce realistic X-ray images, but they are time-consuming. Existing real-time GS-based methods ignore complex X-ray interactions, limiting reconstruction performance. The proposed DDGS model integrates complex X-ray physical interactions into GS to achieve real-time, high-quality X-ray image rendering. This is well-motivated.

**Technical novelty**

The proposed DDGS model introduces novel anisotropic, direction-dependent Gaussians to account for the Compton scattering effect. This is a good idea and technically sound. Moreover, the authors propose a novel Radiodensity-Aware Dual Sampling method for CT volumes.

**Clarity and organization**

This paper is well-written and easy to follow.

**Experimental evaluation**

The authors performed extensive experiments on three different datasets, including comparisons with state-of-the-art (SOTA) methods, ablation studies, and downstream tasks. The results confirm the effectiveness of the DDGS model.

**Weaknesses:**

**Effectiveness of the Radiodensity-Aware Dual Sampling**

From Table 4, the improvements made by the Radiodensity-Aware Dual Sampling are marginal (<0.3 dB), so its effectiveness needs further investigation.

**Lack of CT volumes**

Although the purpose of this work is digitally reconstructed radiographs, showing the CT volume represented by GS is necessary for evaluating the proposed model.

**Visualization results**

The qualitative results shown in Figure 3 are hard to distinguish visually. Please consider showing the error maps between the reconstructions and the ground truth.

**Some typos**

For example, in line 90, “] − 800, −200], “fat” for ] − 200, −100]” should be “[ − 800, −200], “fat” for [ − 200, −100]”.

**Questions:**

See the section of Strengths and Weaknesses, please.

**Limitations:**

The authors mention enough limitations of their work.

---

> ### Author Rebuttal · Authors · 2024-08-06
>
> ### Effectiveness of the radiodensity-aware dual sampling:
> We believe that the accuracy increase compared to the SOTA [7] (i.e., performing uniform sampling) brought by this single contribution (novel radiodensity-based sampling) is still significant.
>
> We should further highlight that our novel CT-specific initialization scheme **significantly improves the convergence of 3DGS models** for DRR rendering, **as shown in the new Table R3** (see rebuttal PDF). E.g., after 500 iterations, our model with the proposed sampling strategy outperforms the one with SOTA even-sampling [7] by +2.06dB, and the one with random-sampling [24] by +1.31dB. The gap slowly reduces as models converge and reach saturation in terms of image quality; but this shows that our contribution facilitates convergence. We will add this insight to the manuscript.
>
> ### CT volume visualization:
> We thank the reviewer for their insightful suggestion. We will add some volumetric visualizations of input CT scans to our qualitative figures. Please refer to the rebuttal Figure R1 (in the attached PDF) to see how they would look. Note that we used the standard medical imaging platform Slicer [r6] to generate these CT visualizations.
>
> ### Error map visualization:
> We thank the reviewer for another valid suggestion. We will add signed error maps to Figure 3 (we find signed error maps to be more insightful than error maps for DRRs, as they highlight where the methods may underestimate or overestimate the overall photon absorption). An example of how these error maps would look **can be found in the new Figure R1** (in the attached PDF), where we compared DRR outputs to real X-ray scans (c.f. discussion with Reviewer `g5Vp`).
>
> ### Typos:
> Thank you, we will further proofread our manuscript before final submission.
>
> For the mentioned typo example, however, we would like to highlight that $]a, b]$ and $(a, b]$ are both valid, equivalent, notations for the left-open interval $a < x \leq b$ (c.f. standard ISO 31-11 for mathematical notations). We understand that the second notation may be more prevalent in our domain and will change the paper accordingly.

---

> > ### Comment · Reviewer_YFhr · 2024-08-13
> >
> > Thank you for the detailed response, which addresses my concerns. I have also reviewed the comments from other reviewers. I believe this paper makes a significant contribution to X-ray imaging. Therefore, I am raising my score from 5 to 6.

---

> > > ### Author Response · Authors · 2024-08-13
> > >
> > > Thank you once again for your thoughtful review and the time that you have dedicated to our work. We truly appreciate your recognition and support of our contributions to DRR rendering. We are especially pleased that our responses not only clarified the points that you raised, but also addressed the concerns of other reviewers from your perspective. We will carefully incorporate your feedback—such as CT visualizations, error maps, and insights on the convergence boost provided by our novel initialization—into the final version of the paper, to further strengthen it.

---

### Official Review · Reviewer_Zg8U · 2024-07-12

**Soundness:** 3
**Presentation:** 2
**Contribution:** 3
**Rating:** 5
**Confidence:** 4

**Summary:**

This manuscript presents a novel method called Direction-Disentangled Gaussian Splatting (DDGS-CT), tailored for balancing realistic X-ray simulation and efficient DRR generation using 3D Gaussian Splatting (3DGS). It addresses the challenges posed by intricate physics computation, which often harms the application of Monte Carlo simulations. The method decomposes the radiosity into isotropic and direction-dependent components to approximate complex anisotropic interactions without runtime simulation. Notably, it achieves significant improvements in PSNR and SSIM on DRR renderings of multiple organs, demonstrating superior performance.

**Strengths:**

1.	Innovative Approach: This manuscript introduces an innovative method of X-ray simulation techniques to balance the efficiency and realism in DRR rendering, which is a significant step forward in the field of medical imaging.
2.	Significant Empirical Improvements: The method substantially improves PSNR and SSIM scores, demonstrating its effectiveness over existing methods.
3.	Detailed Methodological Framework: This manuscript presents a well-structured and comprehensive methodological framework, introducing the disentanglement of isotropic and direction-dependent components and initialization strategy that accounts for the nature of CT data.

**Weaknesses:**

1.  “… we equally split P^dw to initialize both isotropic and direction-dependent Gaussians.” I’m confused about how to split the points set P^dw into disentangled isotropic and anisotropic point sets. The authors should provide a more detailed explanation.
	2. How do the disentangled isotropic and anisotropic 3DGS be rendered? Based on Figure 1, it seems that the two disentangled 3DGSs are integrated to render the output. If so, why should they be disentangled? The authors should explain the motivation for this design.
	3. In the Experiments, the 3D Gaussian-based methods are evaluated, yet the DRR-based methods mentioned above are not validated both in qualitative and quantitative comparison. Can they be added?

**Questions:**

1.	I wonder if it is necessary to disentangle the isotropic and anisotropic point sets since they are both splatted to render the output images. I think if they are disentangled, it would be more reasonable that they are also rendered using different algorithms.
2.	As far as I know, 3DGS is actually an explicit representation of 3D data. I wonder if it is necessary to transform 3D data from an explicit representation (volume) into another explicit representation (3DGS).

**Limitations:**

The authors have adequately discussed their limitations.

---

> ### Author Rebuttal · Authors · 2024-08-06
>
> ### Isotropic vs. anisotropic sets — motivation, initialization, and rendering:
> We hope that the following points will clarify some misunderstandings:
> - We define 2 different functions for the absorption contribution of the 2 Gaussian sets: one function is isotropic to approximate average radio-absorption ($c_i^{\text{iso}}$), and one function is direction-dependent to approximate the contribution of anisotropic Compton scattering to the image ($c_j^{\text{dir}}$).
> - Since we have 2 distinct Gaussian sets, they could indeed be rasterized separately, as done in Figure 4 purely for qualitative ablation. However, the two sets need to be **rendered together to ensure correct simulation**. Otherwise, Gaussians _occluded_ by others belonging to the other set could incorrectly contribute to the ray absorption. Please see Figure R2 in rebuttal PDF for an example: there, point $g_2^{\text{dir}}$ should not contribute to the final pixel value, as it is _occluded_ by $g_2^{\text{iso}}$ (i.e., it absorbs all the remaining energy before the ray can reach $g_2^{\text{dir}}$). This could not be properly simulated if each Gaussian set is rasterized separately.
> - By having 2 sets of Gaussians with distinct yet complementary contribution functions (isotropic vs. direction-dependent functions), our solution can better model DRR imaging. During optimization, each Gaussian set will approximate its respective imaging effect, conditioned by its respective function.
> - We randomly split $\mathbf{P}^{\text{mc}}$ (3D points sampled based on the CT scan's distribution) into 2 sets, each subset contributing to the initialization positions of $g_i^{\text{iso}}$ and $g_j^{\text{dir}}$ (half the 3D points in $\mathbf{P}^{\text{mc}}$ will serve as initialization for $g_i^{\text{iso}}$, along with the entire $\mathbf{P}^{\text{mc}}$ set; and the other half of $\mathbf{P}^{\text{mc}}$ will serve as initialization for $g_j^{\text{dir}}$). After initialization, during 3DGS-based model optimization, the **points in each set of Gaussians can move/split/drop independently** (along with the optimization of their respective covariance/opacity/feature values). Since the optimization of each set (isotropic and anisotropic) is conditioned on their respective differentiable absorption function, they will acquire distinct properties.
>
> We will add more details to the paper and will provide the new figure in the annex of the camera-ready version.
>
> ### Comparison to "DRR-based methods":
> Since our method belongs to the family of _analytical_ DRR rendering methods [7, 11, 29, 30, 24] and to the more recent sub-family of 3DGS-based solutions [7, 24], we evaluated it accordingly. As standard in this domain [5, 7, 11, 12, 33] (see discussion with Reviewer `g5Vp`), we compared on meaningful downstream tasks (CT/X-ray pose estimation and 3D/2D keypoint registration) with corresponding SOTA methods [11, 12, 7] and baselines [17] (note that we did not consider [24], as its contributions w.r.t. [17] are marginal). Additionally, though less common in the domain of analytical DRR rendering (e.g., [7, 11] only compare to a few methods), we also provided an image-quality evaluation over several datasets, comparing to Monte-Carlo (MC) simulation results (used as GT, which is standard in DRR evaluation [5, 7, 11, 33]). Based on a suggestion by Reviewer `g5Vp`, we further provide some quantitative comparison to real projections (see rebuttal PDF).
>
> Therefore, by **combining image-quality evaluation** (comparing to MC and real data) **and downstream-task evaluation on multiple datasets**, against several relevant SOTA analytical methods, we believe that we have been more thorough than most DRR papers.
>
> ### Voxel vs. Gaussian representations:
> The reviewer is correct that both the original CT scan representation (voxel grid) and the proposed 3DGS-based one are explicit. However, as shown in our experiments, our 3DGS representation is **much more compact**. E.g., for the CTPelvic1K data, our model can represent a CT scan with only ~42,625 Gaussians (c.f. Table 1), defined by 19 float values each (3D position/rotation + 3D covariance + opacity + feature vector) and a shared 32-dimensional basis vector $\mathbf{b}$, so $809,907$ float values in total; whereas the original CT scans are each composed of $512\times 512\times 500=93,363,200$ values. E.g., the voxel data can contain large homogeneous regions which can be approximated with few Gaussians, resulting in a **significant compression rate**.
>
> Moreover, the original voxel grids contain raw absorption values, which need further heavy processing to be projected into DRRs (volume traversal and rendering). In comparison, the 3DGS representation is already meant for fast rendering, i.e., facilitating the rasterization process. For the sake of DRR generation, we demonstrate (e.g., in Table 7, against DiffDRR [11, 12] which relies on voxel representation) that 3DGS models are **much lighter, both in terms of forward rendering and backward propagation** (less operations), making them better suited for integration into larger automation systems.

---

> > ### Comment · Reviewer_Zg8U · 2024-08-12
> >
> > After reading the author's response and the other reviews, I would like to keep my initial score.

---

> > > ### Author Response · Authors · 2024-08-13
> > >
> > > We would like to thank the reviewer again for their thoughtful review and for the time dedicated to our submission. We appreciate your recognition of our work and acknowledgement of our contributions ("_innovative approach_", "_significant empirical improvements_", "_detailed methodological framework_"). We will account for your insightful remarks in our final version (additional results, further details on rendering process, and insight into the benefits of GS representation over voxel one for DRR).

---

### Author Rebuttal · Authors · 2024-08-06

We are very grateful to the reviewers `Zg8U`, `YFhr`, `ZGY5`, and `g5Vp` for the constructive feedback, as well as the recognition of our paper's strengths, such as its novelty with regard to existing analytical DRR renderers, its clarity, and its conclusive evaluation on downstream tasks.

In this global response, we further highlight the strengths of our work shared by reviewers and address the only concern brought by more than one reviewer. Other individual questions are answered in our per-reviewer responses, and we are looking forward to further discussing with reviewers.

## Shared Strengths
### 1. Technical Novelty
Reviewers `Zg8U` and `YFhr` agree that our approach is "innovative" [`Zg8U`] / "novel" [`YFhr`], and "**significant step forward in the field of medical imaging**" [`Zg8U`].

Whereas "existing real-time GS-based methods ignore complex X-ray interactions" [`YFhr`], we provide two key contributions, i.e., "introducing the disentanglement of isotropic and direction-dependent components and initialization strategy that accounts for the nature of CT data" [`Zg8U`, `YFhr`, `ZGY5`].

### 2. Significant Empirical Improvement
Reviewers positively responded to our quantitative evaluations and agree on the "effectiveness" [`Zg8U`, `YFhr`] / "excellence" [`ZGY5`] of our algorithm.

Not only does DDGS "**substantially improve PSNR and SSIM** scores [...] over existing methods" [`Zg8U`], but "experiments **combining 2D/3D CT image registration** were also presented" [`g5Vp`]. Reviewer `YFhr` deemed our experiments against "multiple SOTA methods" and "on 3 different datasets" to be "extensive".

### 3. Clear Motivation and Methodology
All four reviewers agree that our motivation / definition of the problem is **clear** [`Zg8U`, `YFhr`, `ZGY5`, `g5Vp`]. Except for `ZGY5`, all reviewers also praised the structure/comprehensiveness of our methodological framework.

Finally, they also agree that we "adequately discussed [our] limitations".

## Shared Question - Comparison to MC Data
Overall, the questions and concerns brought by the reviewers have little overlap, except for the positioning of our work w.r.t. Monte-Carlo simulators, asked by `g5Vp` and implicitly `Zg8U`. We partly clarify below and further answer each reviewer in their respective response.

### MC Applications
We shall better highlight the fundamental differences between Monte-Carlo simulators and analytical DRR renderers, not only in terms of theoretical background but also in terms of target applications/metrics.

DRRs are used either to provide quick visualization to clinicians (e.g., for pre-operative planning or intra-operative guidance) or as an intermediary modality for automation algorithms (e.g., to enable CT/X-ray registration, c.f. our evaluation). Analytical methods have thus been proposed to generate such projections from CT data, navigating the trade-off between speed and photorealism.

On the other hand, MC simulators tackle broader applications (dosimetry validation, photon tracking, etc.) and are usually applied to virtual scenes (e.g., digital twins of scanners, phantoms, geometric primitives, etc.). While some provide the interface to import real CT scans into their virtual scene, the scans have to be manually preprocessed (structure segmentation and material assignment), a tedious and error-prone procedure. Combined with their extremely slow nature (c.f. per-photon simulation of complex physics effects), this makes them ill-suited for online DRR rendering. I.e., their application to DRR rendering is only as a costly source of GT images, not as actual solutions to be compared with.

### Standard Evaluation of DRR Methods
The main goals of DRRs are to provide quick visualization to clinicians (key metrics = speed, visibility) and to integrate larger imaging applications (key metrics: speed, feature-level similarity with real data).

Therefore, prior DRR papers (traditional [11, 29, 30] or GS-based [7, 24]) mostly evaluate on downstream tasks (e.g., pose registration). A few evaluate the image quality compared to other DRR tools ([11] compares to Siddon's and Plastimatch [29]; [7] to other NVS baselines). We found only one [12] that provides a qualitative comparison to some real scans. Quantitative comparison to real data is challenging. Not all imaging parameters are usually provided or accurate enough to create matching DRRs (w.r.t. intensity, CT pose, etc.). It is especially hard to render DRRs aligned with real scans with existing MC tools [2-4, 15]; which is likely why **no DRR paper has performed such evaluation**. Their interfaces and custom coord. conventions are not compatible with pose annotations in public CT/X-ray datasets [13, 25], and their documentation/support is lacking (we tried reaching their authors). Bridging the convention gap between MC and analytical communities would greatly benefit our domain and could be the focus of our next effort.
Note another benefit of DDGS: if real posed scans are available, it could be finetuned on them for higher realism, which is not possible with traditional DRR tools and MC simulators.

### Additional Results
Please refer to our responses to `g5Vp` and `Zg8U` to see **additional results** that we nevertheless successfully gathered.

----
### References mentioned in our responses:
[r1] Mildenhall et al. Nerf: Representing scenes as neural radiance fields for view synthesis. ACM-Comm, 2021.

[r2] Vicini et al. A non-exponential transmittance model for volumetric scene representations. TOG, 2021.

[r3] Garbin et al. Fastnerf: High-fidelity neural rendering at 200fps. ICCV, 2021.

[r4] Tagliasacchi et al. Volume rendering digest (for nerf). arXiv, 2022.

[r5] Zha et al. R2-Gaussian: Rectifying Radiative Gaussian Splatting for Tomographic Reconstruction. arXiv, 2024.

[r6] Fedorov et al. 3D Slicer as an image computing platform for the Quantitative Imaging Network. Magnetic resonance imaging, 2012.

---

### Decision · Program_Chairs · 2024-09-25

**Decision:**

Accept (poster)

**Comment:**

The manuscript introduces the Direction-Disentangled Gaussian Splatting (DDGS-CT) method, aimed at improving the balance between realistic X-ray simulation and efficient DRR generation using 3D Gaussian Splatting (3DGS). The method distinguishes itself by decomposing the radiosity into isotropic and direction-dependent components, which approximates complex anisotropic interactions without runtime simulation. The proposed technique demonstrates significant improvements in PSNR and SSIM for DRR renderings of multiple organs, suggesting superior performance compared to existing methods.

Strengths :
The DDGS-CT method introduces a novel technique for balancing efficiency and realism in DRR rendering, which is a significant step forward in medical imaging.
The method models isotropic and anisotropic contributions via distinct 3D Gaussians to handle complex X-ray interactions like Compton scattering.
It demonstrates substantial improvements in PSNR and SSIM scores, validating its effectiveness over existing methods.

Weaknesses :
The authors have adequately discussed their limitations, but some concerns remain about the robustness and reliability of the method in clinical practice, particularly regarding artifacts and the amplification of errors.

Recommendations:
The authors could discuss the results a little more to show the effectiveness of their method in terms of benefits which sometimes seem marginal. For example, the notion of noise or imprecision in the initial segmentation concerns all methods in any case.

The authors have responded fully and honestly to the reviewers' many comments. They have provided additional information, tables and figures to improve the presentation of their results.